# Neuronal Aβ42 is enriched in small vesicles at the presynaptic side of synapses

Yang Yu[1], Daniel C Jans[2], Bengt Winblad[1], Lars O Tjernberg[1], Sophia Schedin-Weiss[1]

The amyloid β-peptide (Aβ) is a physiological ubiquitously expressed peptide suggested to be involved in synaptic function, long-term potentiation, and memory function. The 42 amino acid-long variant (Aβ42) forms neurotoxic oligomers and amyloid plaques and plays a key role in the loss of synapses and other pathogenic events of Alzheimer disease. Still, the exact localization of Aβ42 in neurons and at synapses has not been reported. Here, we used super-resolution microscopy and show that Aβ42 was present in small vesicles in presynaptic compartments, but not in postsynaptic compartments, in the neurites of hippocampal neurons. Some of these vesicles appeared to lack synaptophysin, indicating that they differ from the synaptic vesicles responsible for neurotransmitter release. The Aβ42-containing vesicles existed in presynapses connected to stubby spines and mushroom spines, and were also present in immature presynapses. These vesicles were scarce in other parts of the neurites, where Aβ42 was instead present in large, around 200–600 nm, vesicular structures. Three-dimensional super-resolution microscopy confirmed that Aβ42 was present in the presynapse and absent in the postsynapse.

## Introduction

Several lines of evidence point towards amyloid β-peptide (Aβ) being a causative agent in Alzheimer disease (AD), but the molecular details behind its role in the initiating events leading to clinical AD are elusive. Pathologically, the AD brain is characterized by extracellular plaques composed of fibrillary Aβ surrounded by activated astrocytes, microglia, and dystrophic neurites. Other pathological hallmarks are accumulation of Aβ in the vasculature, neurofibrillary tangles composed of hyperphosphorylated tau protein, brain atrophy, and loss of neurons and synapses (1). The accumulation of Aβ in AD brain starts several years to decades before clinical symptoms appear (2) and research during the past decades suggest that soluble forms of Aβ rather than Aβ plaques are neurotoxic and correlate with cognitive decline in AD (3).

Different mechanisms behind Aβ toxicity have been suggested. Several studies have shown that oligomeric forms of Aβ can bind to cell surface receptors at neuronal synapses (4, 5), whereas other studies suggest that intracellular Aβ42 is neurotoxic. Analyses of intraneuronal Aβ levels in neurons dissected by laser capture microscopy from postmortem control and AD brain showed that pyramidal neurons in cornu ammonis 1 of AD hippocampus display increased Aβ42 levels and increased Aβ42/Aβ40 ratio in sporadic and familial AD cases (6). The study suggested that there is a correlation between high intracellular Aβ42 levels and vulnerability to AD neuropathology. An electron microscopy study showed that the intracellular accumulation of Aβ42 in human AD brain and a transgenic AD mouse model (Tg2576) occurs predominantly within neuronal processes and an accumulation in synaptic compartments was observed (7). The neurotoxicity of intracellular Aβ is also supported by a study in which autophagy-deficient mice were cross-bred with amyloid precursor protein (APP) transgenic mice (8). Intriguingly, the cross-bred mice show impaired secretion of Aβ, resulting in highly increased intracellular levels of Aβ. This increase was accompanied by severe memory impairment, significantly worse compared with the APP transgenic mice (8).

An increasing amount of studies suggest that Aβ is of importance in normal synaptic function, such as developmental synaptic plasticity (9). Thus, it is important to study the synthesis, localization, and function of Aβ both from a physiological and pathological perspective. Aβ is formed from the APP by the sequential action of two transmembrane aspartyl proteases. BACE-1 first cleaves APP to generate a 99-residue C-terminal fragment (C99), which is subsequently cleaved by the protein complex γ-secretase, generating an APP intracellular domain (AICD) and Aβ of variable lengths (10, 11). The most common form contains 40 amino acids (Aβ40), whereas the longer forms (Aβ42 and Aβ43) are more aggregation-prone and neurotoxic (12, 13). Several studies have shown that APP can be transported along axons and dendrites to nerve terminals (14, 15, 16). Moreover, BACE1 and γ-secretase have been found to be enriched at the neuronal synapse and it has been suggested that Aβ is indeed produced at the synapse (17, 18). Whether or not neuronal Aβ is secreted in an activity-dependent manner has been debated, and a recent study suggested that the

[1]Center for Alzheimer Research, Division of Neurogeriatrics, Department of Neurobiology, Care Sciences, and Society, Karolinska Institutet, Huddinge, Sweden    [2]Science for Life Laboratory, Department of Applied Physics, Royal Institute of Technology, Stockholm, Sweden

Correspondence: sophia.schedin.weiss@ki.se

release of Aβ is mostly constituent and not significantly correlated to the activity-dependent release of glutamate (19). However, neither the physiological importance nor the mechanism by which this occurs has been clarified.

Here, we asked the following questions: Is Aβ42 found all over neurites or enriched at the synapse? Is it found on the pre- or postsynaptic side or both? Which intracellular structures/organelles is it associated with? Confocal microscopy with a best possible resolution of around 200 nm is not sufficient to resolve most synaptic structures. For instance, synaptic vesicles are 40–50 nm in diameter and the synaptic cleft is only 20–25 nm wide. Therefore, we used super-resolution microscopy to study the localization of endogenously derived Aβ42 in the neurites of primary hippocampal neurons. Intriguingly, we were able to resolve different types of fine structures of intracellular Aβ42 that cannot be resolved by confocal microscopy, revealing novel information about the synaptic localization of Aβ42 that may be crucial for understanding its physiological function and the mechanism behind its neurotoxicity.

# Results and Discussion

This study was performed to determine the precise localization of endogenous Aβ42 in neurites and synapses in cultured hippo-campal neurons by super-resolution microscopy. It has previously been shown that intraneuronal levels of Aβ42 correlate with neuropathology in AD (20, 21). There is also a correlation between synaptic degeneration and development of AD, and Aβ42 has been shown to be synaptotoxic (4, 16, 22, 23). Because the synaptic cleft is only 20–25 nm wide (24) and the best possible resolution obtained by confocal microscopy is around 200 nm, super-resolution mi-croscopy is required to determine the synaptic localization of macromolecules with sufficient resolution and precision. A recent study using super-resolution microscopy to investigate the local-ization of the Aβ-producing protease γ-secretase in the neurites showed that this enzyme is enriched at the synapse and that it is present at both the pre- and the postsynaptic side (18). Because γ-secretase can process around 100 substrates besides APP, it is important to determine at which subcellular localizations it can perform its various activities. Here, we studied the localization of Aβ42 in the neurites by using an Aβ42-specific antibody. This an-tibody has been thoroughly characterized previously (25, 26). It was further validated here by Western blotting (WB) (Fig S1), supporting the notion that this Aβ42-specific antibody targets the free C-ter-minus of Aβ42 and has no cross-reactivity with Aβ40 or full-length APP. The presynaptic marker synaptophysin and the postsynaptic marker PSD95 were visualized by antibodies previously shown to be suitable for stimulated emission depletion (STED) and stochastic optical reconstruction microscopy (STORM) imaging of hippo-campal neurons (18). The structure of the neurites was visualized using phalloidin, which binds to filamentous actin. STED experi-ments were set up with two STED channels, one for Aβ42 and one for either of the two synaptic markers (synaptophysin or PSD95). The same depletion doughnut was used in both STED channels to ensure perfect colocalization between these two channels. The actin pattern was imaged in a third, confocal channel throughout

this study. The synaptic localization of Aβ42 was further confirmed by two-color dSTORM experiments.

The fluorescence labeling of Aβ42, synaptic markers, and fila-mentous actin was first optimized and analyzed by confocal mi-croscopy. The culturing of mouse primary hippocampal neurons for 21 d in vitro results in an elaborate network of axons and dendrites with well-developed specialized synaptic contacts, including dense PSD clusters in the spine heads and enrichment of synaptophysin-containing synaptic vesicles on the opposing, presynaptic side (18, 27). The specificity of the fluorescence labeling was analyzed by using negative controls, lacking one of the primary antibodies, to confirm that there was no cross-talk between the channels and that the secondary antibodies did not cross-react (Fig S2). In agree-ment with previous studies, Aβ42 staining was present in the perinuclear region and in the neurites (27).

## STED resolves structures and localization of Aβ42 that cannot be resolved by confocal microscopy

Confocal images of the network of the neurites in samples stained for Aβ42, synaptophysin, and filamentous actin showed an accu-mulation of Aβ42 at the synapse, but it was not possible to dis-tinguish whether Aβ42 was localized to the pre- or postsynapses (Fig 1A). Imaging the same regions using STED microscopy we were able resolve the fine structures in the neurites (Fig 1B). Close-ups of the STED images (Fig 1B, middle and right panels) demonstrated that synaptophysin and Aβ42 are interspersed within the same regions in the neurites, suggesting that Aβ42 is present at the presynapse. Because our previous study showed that axonal γ-secretase is enriched in the presynapse (18), we propose that this pool of intracellular endogenous Aβ42 is synthesized in the pre-synaptic region.

A comparison between confocal and STED microscopy was also performed for samples stained for the postsynaptic marker PSD95, Aβ42, and filamentous actin (Fig 1C and D). Confocal microscopy showed that Aβ42 clusters were located close to PSD95-containing spine heads (Fig 1C). With STED microscopy we were able to reveal that such clusters contain small vesicles adjacent to, but not colocalizing with, the postsynaptic spine heads (Fig 1D), in line with a presynaptic localization. At a nonsynaptic part of a neurite, we noted a different type of Aβ42 structure, which only contained Aβ42 on the rim and was much larger than the vesicles found at the synapse (Fig 1D and see below for more details). Notably, these different Aβ42-containing structures, small vesicles at the synapse and larger nonsynaptic bodies, could only be resolved and dif-ferentiated by using STED. Our findings thus demonstrate that super-resolution microscopy is required both to determine the synaptic localization of Aβ42 and to distinguish between different types of Aβ42 structures in the neuronal compartments.

Although the actin staining is present both in dendrites and in most of the axons, the staining in dendrites is considerably denser, particularly at the spine heads, whereas the axonal staining is weaker (18). However, by adapting contrast settings in the images, it is possible to optimize visualization either of the densely stained dendrites (Fig 2A and B, left panels) or, with increased brightness, the axons (Fig 2A and B, right panels). As in previous studies, the

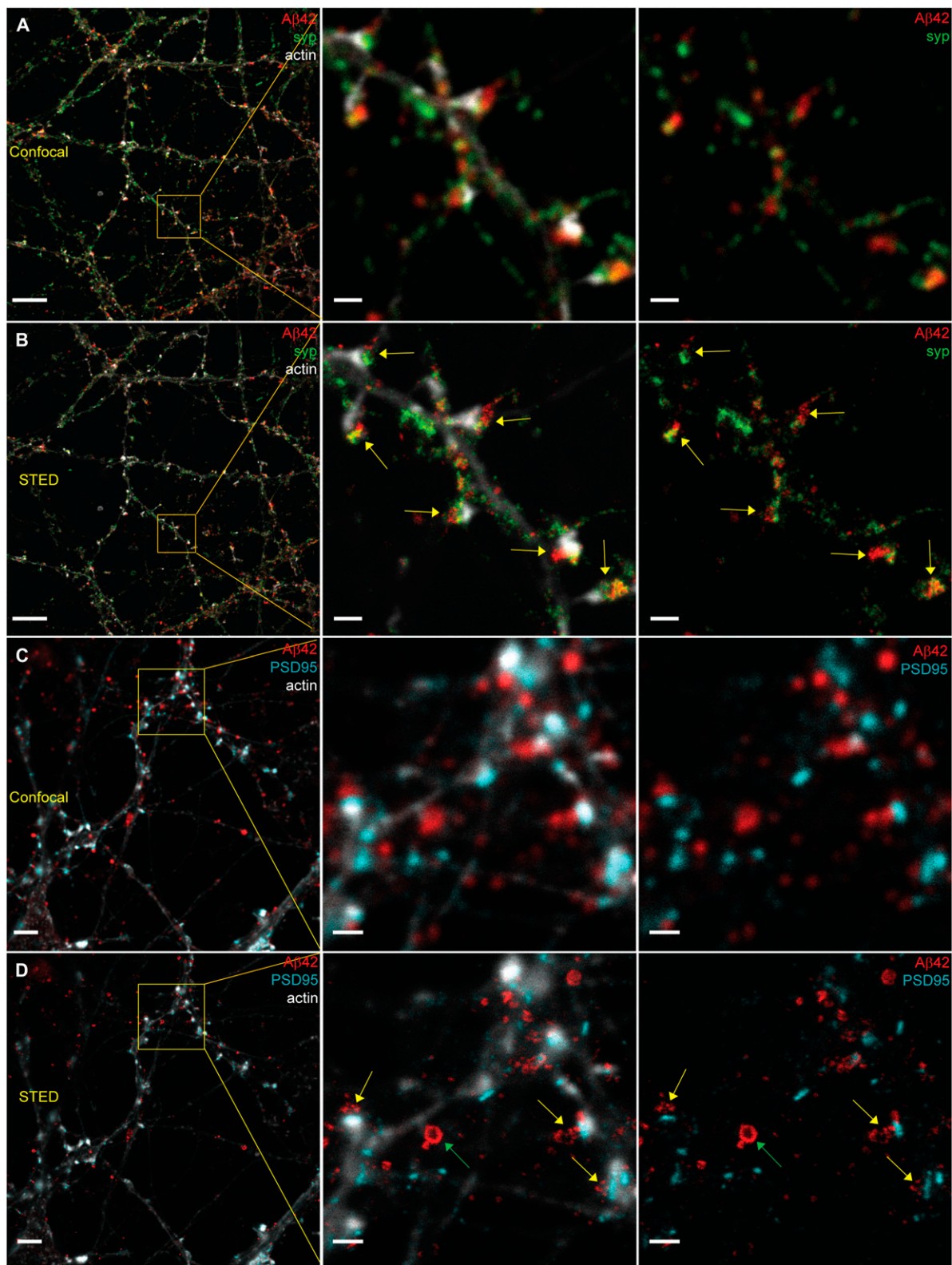

**Figure 1. Comparison of confocal and STED images for studying the synaptic localization of Aβ42 in hippocampal neurons.**
**(A, B)** Hippocampal neurons stained to visualize Aβ42 (red), synaptophysin (green), and actin (light grey) were imaged by confocal microscopy (A) and STED microscopy (B). Yellow arrows point at enrichment of Aβ42 at the synapse. Left scale bars: 10 μm. Other scale bars: 1 μm. **(C, D)** Hippocampal neurons subjected to immunocytochemistry to visualize Aβ42 (red), PSD95 (cyan), and actin (light grey) were imaged by confocal microscopy (C) and STED microscopy (D). Yellow arrows point at enrichment of Aβ42 at the synapse. Green arrow points at an Aβ42-stained vesicle where Aβ42 is stained only at the rim of the vesicle. Left scale bars: 4 μm. Other scale bars: 1 μm.

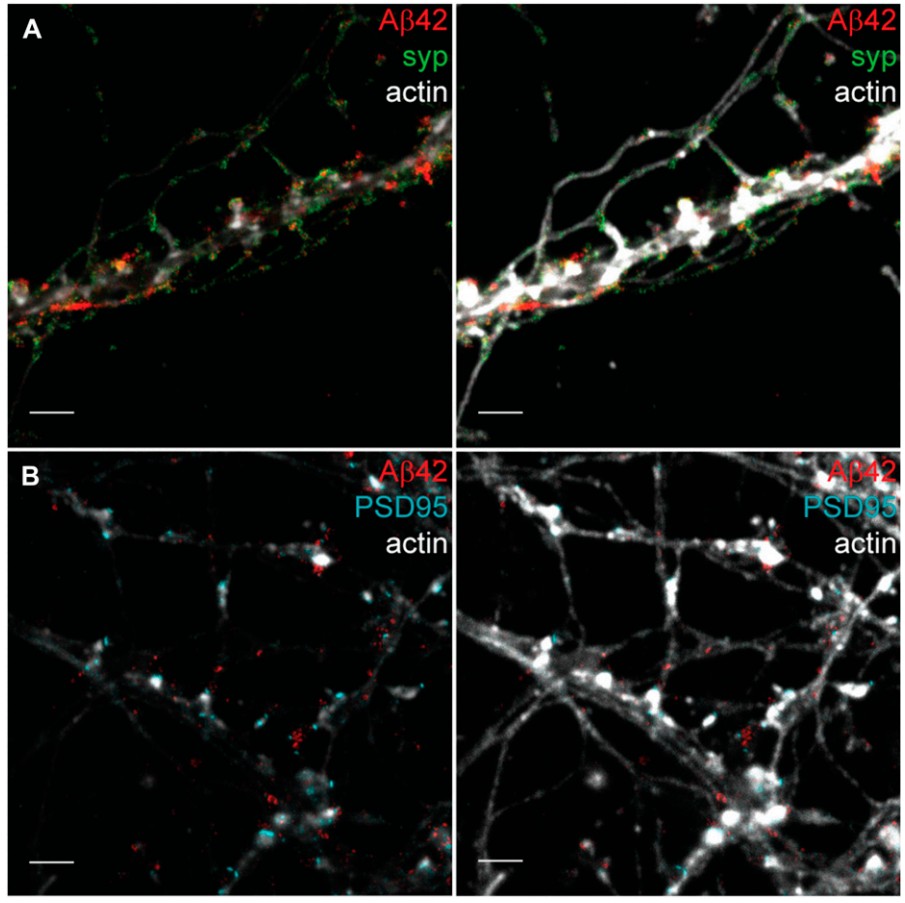

**Figure 2. STED images of Aβ42 demonstrating the intricate axonal network surrounding dendrites in the 21-d in vitro hippocampal neurons.**
**(A)** STED image showing Aβ42 (red) and synaptophysin (green) combined with a confocal channel for actin (light grey). The left image has a lower brightness in the actin channel than the right image. Note that the axonal staining appears for many axons only in the right image with increased brightness. This image was further zoomed-in in Fig 3A to focus on the synapses. Scale bars: 2 μm. **(B)** STED image showing Aβ42 (red) and PSD95 (cyan) combined with a confocal channel for actin (light grey). The left image has a lower brightness in the actin channel than in the right channel. Note the intensive axon network both between and along the neurites. Scale bars: 2 μm.

dendrites have extensive axonal networks along with as well as between them (18).

### Analysis of the distribution of Aβ42 at the pre- and postsynaptic sides by STED

Analyzing a large number of synapses from different areas of the dish, we found that most of the Aβ42 staining in the neurites was present within the same region as the synaptophysin clusters. Interestingly, both Aβ42 and synaptophysin appeared to be present in vesicular structures of similar size, and to some extent colocalized (Figs 3A and B, and S3A and B). The appearance of the synaptophysin staining is in agreement with previous electron microscopy studies showing that synaptic vesicles are uniform in size with a diameter of around 40–50 nm (28). It is highly intriguing that we now observed that some vesicles of similar size as synaptic vesicles stained for Aβ42 but not synaptophysin. Because colocalization of Aβ42 and synaptophysin was observed in some areas, it is unlikely that the absence of synaptophysin staining in these vesicles is due to steric hindrance of the antibody but rather indicates the presence of Aβ42-containing vesicles that lack synaptophysin. To further test this hypothesis, we compared the samples stained for Aβ42 and synaptophysin with samples stained for synaptophysin and another synaptic vesicle protein—synaptobrevin. In the presynaptic regions of the Aβ42/synaptophysin samples, 19% of

the stained pixels were positive for only Aβ42, 36% were positive for both Aβ42 and synaptophysin, and 45% were positive for only synaptophysin (Fig S4). Thus, 35% of the Aβ42-stained pixels did not colocalize with synaptophysin and 56% of the synaptophysin-stained pixels did not colocalize with Aβ42 (Fig S4). Doing the same operation for the synaptobrevin/synaptophysin samples, we found that 15% of the stained pixels were positive for only synaptobrevin, 52% were positive for both synaptobrevin and synaptophysin, and 33% were positive for only synaptophysin. Thus, only 22% of the synaptobrevin-positive pixels did not colocalize with synaptophysin, and 39% of the synaptophysin-positive pixels did not colocalize with synaptobrevin. The synaptobrevin/synaptophysin colocalization was somewhat lower than we expected, possibly because of labeling efficacies lower than 100%. Another potential explanation is that the size of the primary–secondary antibody complex may provide a detection area somewhat larger than the actual vesicle size. Alternatively, the vesicle pool could be more heterogeneous than expected. Still, the lower extent of colocalization of Aβ42/synaptophysin compared with synaptobrevin/synaptophysin in the presynaptic region supports our notion that part of the Aβ42-containing vesicles may be distinct from the synaptic vesicles responsible for neurotransmitter release. Previous work has shown that Aβ42 release can occur independently of glutamate release (19), supporting our notion that a fraction of Aβ42 is present in vesicles that differ from the vesicles responsible for transmitter release.

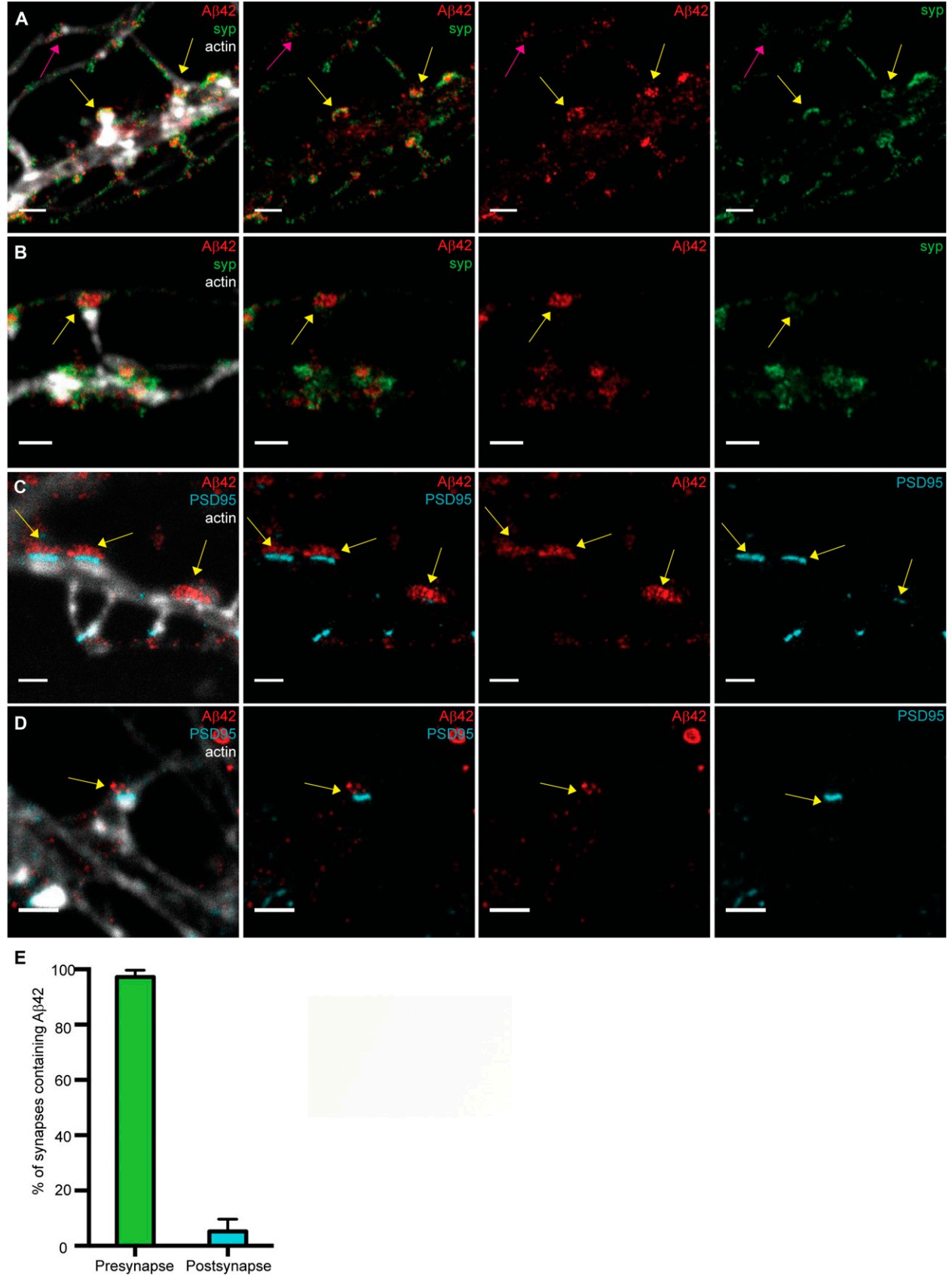

Aβ42 localization at the neuronal synapse   Yu et al.                    https://doi.org/10.26508/lsa.201800028    vol 1 | no 3 | e201800028    **5 of 11**

The appearance of vesicles that contain Aβ42 and/or synaptophysin was observed both in immature presynapses that have not yet formed a contact with a dendritic spine (Fig 3A) and in presynapses that are in contact with dendritic spine heads (Fig 3B). Notably, our previous super-resolution microscopy study showed that the enzyme complex γ-secretase, which is required for the generation of Aβ42, is also present in both immature and mature synapses (18). Together, these findings imply that Aβ42 and/or γ-secretase have a role(s) in synaptic formation and maintenance.

PSD95 imaging by STED beautifully outlined PSD clusters at the top of the spine heads and these clusters, in contrast to synaptophysin clusters, showed no or only little overlap with Aβ42 (Figs 3C and D, and S3C and D). A clear enrichment close to the PSD clusters was observed, on the other side of the synapse, again supporting the presynaptic localization of Aβ42. This presynaptic clustering of Aβ42 vesicles opposite to PSD clusters was obvious both for stubby spines (Fig 3C) and mushroom spines (Fig 3C and D). Quantification of individual synapses stained for Aβ42, phalloidin, and synaptophysin or PSD95 showed that the majority (97 ± 2.5%) of presynapses were positive for Aβ42 vesicles, whereas only very few of the postsynapses (5.2 ± 4.5%) were positive for Aβ42 staining (Fig 3E). The latter could be because of a small extent of false positive results if some of the dendritic spines were not perfectly aligned perpendicular to the optical axis, i.e., because of the lower resolution along the z-axis.

### Aβ42 is present in large vesicles in nonsynaptic regions of the neurites

The occurrence of larger Aβ42-containing structures (Fig 1D), with a diameter ranging between 200 and 600 nm (Fig S5) and enriched in Aβ42 in the outer portions, were further investigated. Although the staining was intense at the rim, Aβ42 is most likely enriched in the interior of the vesicle, close to the membrane. In contrast to the smaller vesicles, these vesicles were located on or close to neurites in regions lacking spine heads, synaptophysin, or PSD95 clusters (Fig 4A and B), indicating that they are not present at the synapse. Interestingly, a previous electron microscopy study showed that Aβ42 is enriched at the membrane of multivesicular bodies (MVBs) in neurons both in human AD brain and in brain from mice and rats, especially in pyramidal neurons from the cortex and hippocampus, which are most prone to develop AD neuropathology (7). Furthermore, it has also been shown that Aβ can be produced in a macroautophagy-dependent pathway involving autophagosomes and late autophagic vacuoles, which are multivesicular entities with a diameter of around 500 nm (29). Because intraneuronal Aβ42 can be directly derived by endogenous production or secreted and taken up by endocytosis of external Aβ42, it is possible that the sources of Aβ42 in MVBs and presynaptic vesicles are of different origin.

### Analysis of the distribution of Aβ42 at the pre- and postsynaptic sides by dSTORM

The size of synaptic structures and vesicles makes diffraction unlimited super-resolution microscopy indispensable for investigating synaptic localizations of proteins. All diffraction unlimited techniques rely on switching between two discernible states, and two classes of diffraction unlimited super-resolution techniques can be distinguished: spatially targeted methods such as STED and spatially stochastic techniques such as, for instance, photoactivated localization microscopy and STORM (30, 31). All these methods can overcome the diffraction barrier substantially. So, we asked if the synaptic localization of Aβ42 can also be seen using a spatially stochastic super-resolution technique. Here, we applied dSTORM (32, 33), using the two photo-switchable fluorophores, Alexa Fluor 647 and Atto 488, as reporters for Aβ42 and the synaptic marker, respectively. In agreement with the STED images, Aβ42 was dispersed within the same region as synaptophysin (Fig 5A) but not as PSD95 (Fig 5B), supporting our notion that Aβ42 is localized at the presynapse in pyramidal neurons. Importantly, there was no or very little direct overlap between Aβ42 and synaptophysin staining. These findings show that Aβ42 localization can be analyzed also by using spatially stochastic approaches and support our suggestion that Aβ42 is partially present in vesicles different from those containing synaptophysin.

### 3D STED confirms the presynaptic localization of Aβ42

The confocality of a STED microscope and the easy implementation of additional (confocal) reference channels make it particularly suitable also for analyzing complex structures such as neuronal cultures. To analyze pre- or postsynaptic protein localizations, the positioning of the synapse with respect to the optical axis of the microscope has to be considered. Conventional 2D STED using a vortex phase plate provides very high lateral resolution but the axial resolution is still unaltered confocal, which means that structures that are aligned along this axis cannot be resolved beyond the diffraction limit. Therefore, we focused in the previous part on synapses that are aligned perpendicular to the optical axis. To analyze the localization of Aβ42 in a 3D context, we used 3D STED, providing super-resolution in all directions. The presynaptic localization of Aβ42 could be confirmed using 3D STED. We found Aβ42 partially colocalizing with synaptophysin (Video 1), whereas the lack of overlap between the PSD95 and Aβ42 staining (Video 2) was apparent in all three dimensions, confirming our 2D STED observations.

In summary, by using super-resolution microscopy we could, in hippocampal neurons, show that endogenous Aβ42 is present at

---

**Figure 3. STED imaging of Aβ42 focusing on the pre- or postsynaptic regions.**
**(A, B)** Zoomed STED images of synapses showing Aβ42 (red) and the presynaptic marker synaptophysin (green). Neuronal structure is shown by actin in a confocal channel (light grey) merged to the STED channels in the left panel. The staining of synaptophysin and Aβ42 at presynaptic boutons connected to the postsynaptic spines are shown by the yellow arrows. The staining of synaptophysin and Aβ42 at "free" boutons that do not connect to the postsynaptic spine is shown by a pink arrow. Scale bars: 1 μm. **(C, D)** Zoomed STED images of synapses showing Aβ42 (red) and the postsynaptic marker PSD95 (cyan). Neuronal structure is shown by actin staining in a confocal channel (light grey) merged to the STED channels in the left panel. Yellow arrows point at Aβ42 vesicles in axonal regions opposite to PSD95-containing stubby spines in (A) and to Aβ42 vesicles in axonal regions opposite to a mushroom spine in (B). Scale bars: 1 μm. **(E)** Quantification of Aβ42 staining in pre- and postsynapses. The staining of Aβ42 combined with the presynaptic marker synaptophysin (green bar) or postsynaptic marker PSD95 (cyan bar) was quantified from STED data. Graphpad prism was used to prepare graphs showing the proportion ± SD of presynapses (n = 73) and postsynapses (n = 73) that were positively stained by Aβ42.

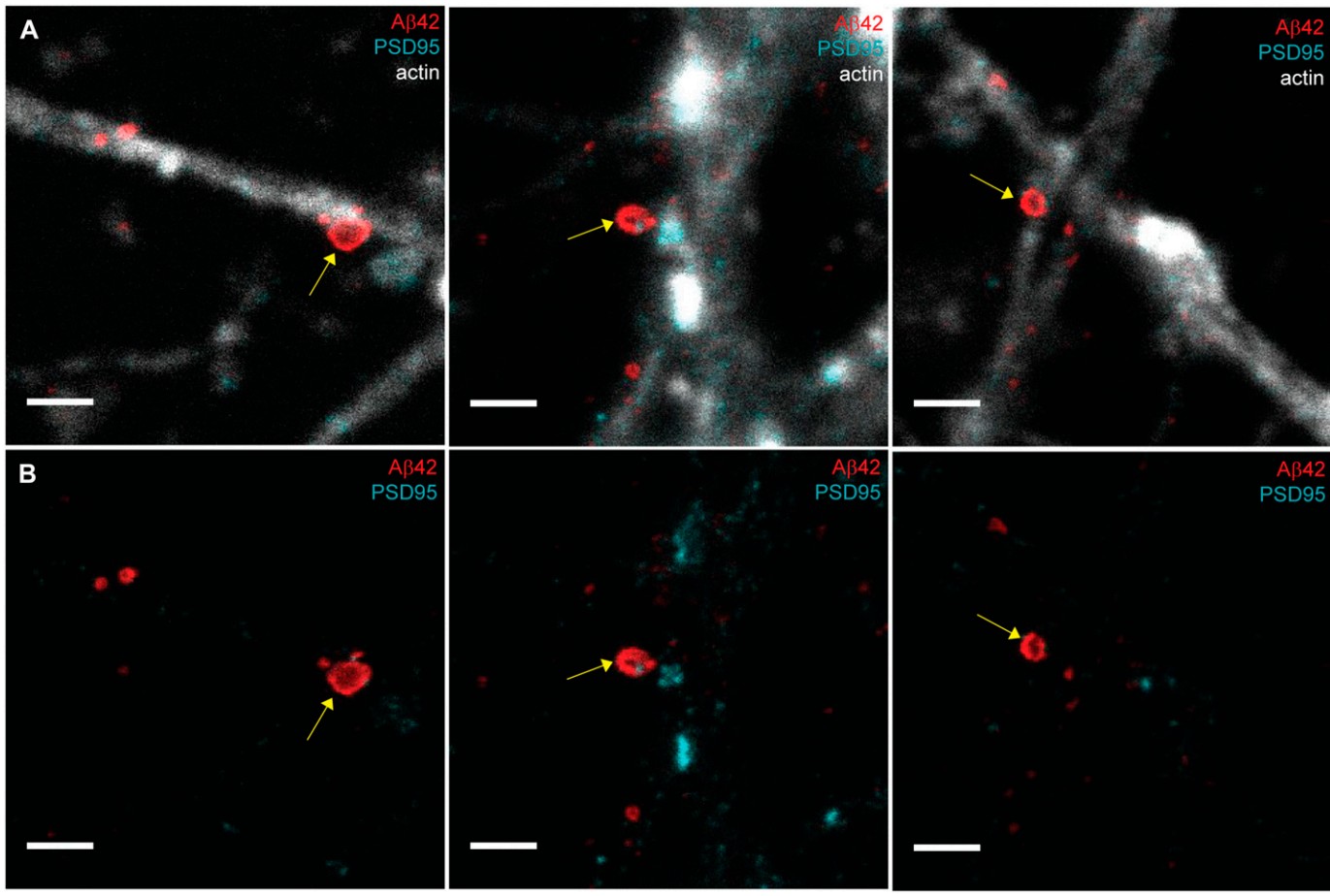

**Figure 4. Aβ42 staining at nonsynaptic neuritic regions.**
Zoomed STED images of Aβ42 (red), PSD95 (cyan), and actin (light grey) at nonsynaptic neuritic regions (A) and only the Aβ42 and PSD95 channel (B). Yellow arrows show Aβ42 vesicles which are hollow inside. The diameter from left to right is 520, 480, and 490 nm, respectively. Images were zoomed from Fig S3. Scale bars: 1 μm.

the pre- but not postsynaptic side. Interestingly, presynaptic Aβ42 was localized to vesicles of similar size as synaptic vesicles, some of which lacked synapthophysin, suggesting that a fraction of Aβ42 is present in a previously non-characterized vesicle in the pre-synapse. Intriguingly, these data may explain the controversy regarding the dependency of Aβ secretion on synaptic activity. Moreover, we could also resolve larger, nonsynaptic structures in which Aβ42 showed a donut-like appearance, in line with the presence of Aβ42 in the outer membrane of MVB. Thus, these findings provide crucial information on the subcellular details and the molecular mechanisms behind the role of Aβ42 in normal synaptic function and in AD pathogenesis.

# Materials and Methods

## Mice

The C57BL6 mice used in this study for preparing hippocampal neurons and mouse brain homogenate were treated according to the Karolinska Institutet's and the national guidelines. Embryonic mice brains (E16.5) were used for the dissection of hippocampi. The brains of the female mice carrying the embryos were used for WB. After dissection, these brains were homogenized in a buffer containing 20 mM Hepes, 50 mM KCl, 2 mM EGTA, and complete protease inhibitor cocktail (Roche), at pH 7.5. No experiments were performed on live animals. The study was approved by the animal research ethical committee in southern Stockholm. Animals were euthanized by cervical dislocation.

## Hippocampal cell culture

Mouse hippocampal neurons were cultured in 35-mm glass bottom culture dishes (P35-G-1.5-10-C; MatTek) with 2% B27 (Thermo Fisher Scientific) and 1% L-glutamine (Gibco) in Neurobasal medium (Gibco), essentially as described previously (34, 35). Briefly, the dishes were coated with poly-D-lysine (P7405; Sigma) at least 1 d before the dissection. Hippocampi and cortices were dissected and single cells were triturated by repeated pipetting of the tissue in cell medium. 7,500 hippocampal cells (with 125 μl medium) were seeded on glass in the middle of the plate, and 150,000 cortical cells (with 500 μl medium) were seeded as support cells on the surrounding plastic (to avoid direct contact between the cell types). The cells were kept in the incubator (37°C, 5% $CO_2$) for 2 h and then

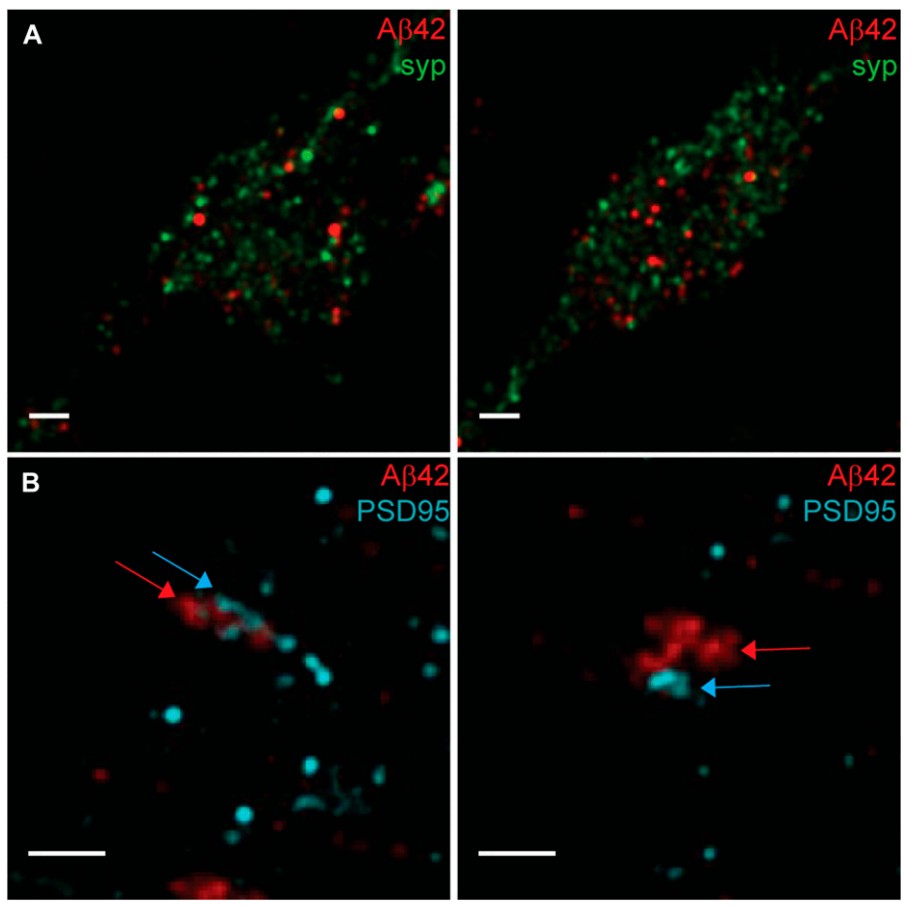

**Figure 5.  Two-color dSTORM images of Aβ42 and pre- or postsynaptic markers.**
**(A)** dSTORM images show Aβ42 (red) and synaptophysin (green). Two different presynaptic regions from the same image are shown in the left and right image. Note that the Aβ42 staining appears as vesicles that are interspersed in the same region as synaptophysin. Scale bars: 500 nm. **(B)** STORM images show Aβ42 (red) and PSD95 (cyan). Two different postsynaptic regions from the same image are shown in the left and right image. Note the proximity between Aβ42 and PSD95, in agreement with a synaptic cleft between the two stains. The somewhat overlap in the left panel is probably related to the fact that the synapse is not in the plane of the image. Scale bars: 500 nm.

2.5 ml fresh medium was added. After 21 d, the cells were fixed with 4% formaldehyde (252549; Sigma) for 10 min at RT and stored in Dulbecco's phosphate-buffered saline (DPBS) (Gibco) at 4°C for immunocytochemistry staining.

### Antibodies and reagents

A rabbit antibody specific for the C-terminal neo-epitope of Aβ42 (anti-AβC42) was used for immunofluorescence labeling of Aβ42 throughout these studies. This affinity-purified antibody, targeting the six C-terminal residues of Aβ42, has previously been shown not to cross-react with either Aβ40 or full-length APP (25). To further validate the antibody, WB blotting was performed and compared with mouse antibody 4G8 (BioLegend), which is reactive to amino acid residues 17–24 of the Aβ sequence within APP. IRDye 800CW (LI-COR) and IRDye 680RD secondary antibodies (LI-COR) were used for WB.

Mouse anti-PSD95 IgG, here denoted anti-PSD95 (Ab2723; Abcam), was used as a postsynaptic marker whereas mouse anti-synaptophysin IgG, here denoted anti-synaptophysin (SP15; Enzo Life Sciences), was used as a presynaptic marker. Mouse anti-synaptobrevin IgG, here denoted anti-synaptobrevin (104203; Synaptic Systems), was used as another presynaptic marker. All of these antibodies have previously been shown to be suitable for confocal, STED, and STORM experiments (18). The Atto 488–labeled secondary antibody used for dSTORM was prepared by the conjugation of

NHS-Atto 488 (ATTO-TEC) to donkey anti-mouse IgG (A16019; Life Technologies). Labeling was performed essentially as described previously (18) by incubating Atto 488-NHS (0.02 mg/tube) dissolved in 10 μl anhydrous DMSO, with 50 μl donkey anti-mouse IgG (1.25 mg/ml in PBS), 6 μl of 1 M NaHCO$_3$, and 1.5 μl Atto 488-NHS for for 30 min at RT, followed by purification using size exclusion chromatography on NAP-5 columns (GE Healthcare). Fluorescently labeled commercial secondary antibodies used were anti-rabbit IgG Fab Alexa Fluor 647 (Invitrogen), anti-rabbit IgG Abberrior STAR 635P, and anti-mouse IgG Alexa Flour 594 (Invitrogen). The neuronal structure was visualized by staining the actin cytoskeleton with Alexa Flour 488–phalloidin (Invitrogen).

The imaging buffer for STORM experiments containing 0.124 M cysteamine (Sigma), 44.8 mM HCl, 8.6% glucose, 1.08 mg/ml glucose oxidase from *Aspergillus niger* (Sigma), and 0.0773 mg/ml catalase from bovine liver (Sigma) in PBS was mixed directly before imaging and used for no longer than 1 h. Fresh stock solutions (1 M cysteamine in 360 mM HCl, 10% Glucose in PBS, 70 mg/ml glucose oxidase in PBS, and 20 mg/ml catalase in PBS) were prepared the day before imaging and stored at 4°C.

### WB

Purified recombinant human Aβ42 and mouse brain homogenate samples were diluted with LI-COR sample loading buffer and

separated on 4%–12% Bis–Tris gel (Invitrogen) in MES SDS running buffer (Invitrogen). Proteins were transferred onto nitrocellulose membranes (GE Healthcare). Membranes were blocked by Odyssey blocking buffer (927-50000; LI-COR) for 1 h at RT, followed by incubation with Aβ42-specific antibody and 4G8 primary antibody overnight. The membranes were washed with 0.1% Tween-TBS for 3 × 5 min, incubated with anti-mouse and anti-rabbit LI-COR secondary antibodies for 1 h at RT, washed again, and followed by imaging in the Odyssey imaging system. The system can detect two protein targets that have been incubated with two different secondary antibodies labeled with spectrally distinct near-infrared fluorophores. The results can be shown as a merge image and as separate 700- and 800-nm channel images in pseudo-color or grayscale.

### Sample preparation for STED and dSTORM

Fixed cells were permeabilized in 0.4% CHAPSO (Merck Millipore) in DPBS for 10 min at RT and blocked with 10% normal goat serum (NGS; Invitrogen) for 15 min at RT. Dishes were subsequently incubated with the primary antibodies in 3%–5% NGS in DPBS overnight at 4°C.

For the preparation of samples for STED imaging, anti-Aβ42 (1:100) was used in combination with either anti-PSD95 diluted 1:200 or mouse anti-synaptophysin diluted 1:200 followed by washing with PBS 3 × 5 min. For colocalization studies, anti-synaptophysin (1:200) was also used in combination with anti-synaptobrevin (1:200). Incubation with the secondary antibodies, anti-rabbit Abberior Star 635P (1:200), anti-mouse Alexa Fluor 594 (1:200), and Alexa Fluor 488–phalloidin (1:100), was subsequently performed for 2–3 h, followed by washing with 0.1% tween in PBS for 3 × 10 min and PBS for 1 × 5 min. The samples were postfixed with 3% formalin and 0.1% glutaraldehyde in DPBS for 10 min at RT. After a final washing step for 3 × 5 min with DPBS and 1 × 5 min with water, the samples were mounted with ProLong gold antifade reagent (Life Technologies) and stored in the fridge for staining. The samples were first analyzed by using a Nikon point scanning confocal A1+Si inverted microscope using a 100× (NA: 1.55) oil immersion objective with an image size of 1,024 × 1,024 pixels. Excitation lasers were 405, 488, 561, and 640 nm.

For the preparation of samples for dSTORM imaging, anti-Aβ42 (1:200) was used in combination with either anti-PSD95 diluted 1:200 or mouse anti-synaptophysin diluted 1:500.

After the incubation, the dishes were washed with PBS for 3 × 5 min. Subsequently, they were incubated with secondary antibodies, anti-rabbit IgG Fab 647 diluted 1:200 (Invitrogen) and anti-mouse IgG Atto 488 diluted 1:200, in 5% NGS in DPBS for 2–3 h followed by washing with 0.1% tween in PBS for 3 × 5 min and PBS for another 3 × 5 min. The samples were subsequently postfixed with 3% formalin and 0.1% glutaraldehyde (Sigma) in DPBS for 10 min at RT. The final washing step was performed for 3 × 5 min with DPBS and 1 × 5 min with water, followed by storage in DPBS until imaging.

### STED imaging

STED imaging was performed on a Leica TCS SP8 STED 3× (Leica Microsystems) equipped with an HC PL APO 100×/1.40 Oil STED WHITE objective. In brief, the fluorophores Alexa Fluor 594 and Abberior Star 635P were excited at 598 and 653 nm, respectively, and STED was performed at 775 nm for both color channels. The dye Alexa Fluor 488 was excited at 499 nm and recorded confocally. The channels were recorded sequentially. All images except the movies are raw data. No image processing, except for contrast stretching, was applied. The 3D STED data were deconvolved using Huygens deconvolution software (Scientific Volume Imaging) and rendered and animated using Imaris (Bitplane).

### dSTORM imaging

dSTORM imaging was performed using an Elyra PS.1 microscope (Carl Zeiss Microscopy) equipped with a Plan-Apochromat 100×/1.46 oil objective and a liquid cooled EMCCD camera (Andor Technology). Imaging was carried out in MEA imaging buffer as previously described (36). In short, fresh stock solutions (1 M cysteamine in 360 mM HCl, 10% glucose in PBS, 70 mg/ml glucose oxidase in PBS, and 20 mg/ml catalase in PBS) were prepared the day before imaging and stored at 4°C and mixed directly before imaging to final concentrations of 0.124 M cysteamine (Sigma), 44.8 mM HCl, 8.6% glucose, 1.08 mg/ml glucose oxidase from *Aspergillus niger* (Sigma), and 0.0773 mg/ml catalase from bovine liver (Sigma) in PBS. Imaging was performed in 12.8 × 12.8-μm areas in an inclined total internal reflection fluorescence microscope mode (37). Single molecule fluorescence detection on the EMCCD camera was acquired with 100 × 100-nm pixel size, 20-ms Exposure time, and 100 Gain. 20,000 image frames were acquired for each channel. Both channels were imaged sequentially in 500 frame sequences and the appropriate filters and lasers for each dye were used (642 nm for Alexa Fluor 647 and 488 nm for Atto 488). The images were analyzed with the ImageJ plugin SMLocalizer (38).

### Quantification of Aβ42 in the synapse

The relative presence of Aβ42 in the pre- versus postsynaptic side of the synapse was quantified from STED images of Aβ42, actin, and synaptophysin or PSD95 staining. The synapses to be included were selected based on the staining of actin and the synaptic markers (synaptophysin for the presynaptic side and PSD95 for the postsynaptic side). Criteria used for selecting synapses were as follows: (i) distinct spine structure (revealed by actin staining) in the plane of the slide; (ii) having a connection to a presynaptic bouton on the side; (iii) not overlapping with other dendrite or axon; and (iv) presence of a PSD cluster in the spine head (for postsynaptic localization) or synaptophysin cluster in a bouton (for presynaptic localization). The selected synapses were marked in Photoshop by yellow circles (Figs S6 and S7). A total of 73 pre- or postsynapses were selected from three to four different images. Graphs showing the proportion ± SD of pre- and postsynapses with Aβ42 staining were prepared using Graphpad prism.

### Comparison of colocalization of Aβ42/synaptophysin with synaptobrevin/synaptophysin

The STED images of Aβ42/synaptophysin and synaptobrevin/synaptobrevin were analyzed using ImageJ. Presynaptic regions were manually selected as regions of interest. The threshold was

set by using the Otsu algorithm, and colocalization was analyzed based on the area of staining. A total of 50 presynapses were selected for each staining combination and the percentage of Aβ42 or synaptobrevin colocalized with synaptophysin was displayed using Venn charts. Total stained pixels in each chart was set to 100% (Fig S4).

## Supplementary Information

## Acknowledgements

We acknowledge the Advanced Light Microscopy facility at Science for Life Laboratory included in the National Microscopy Infrastructure (VR-RFI 2016-00968) for excellent support with super-resolution fluorescence imaging. L Tjernberg acknowledges support from the Swedish Alzheimer Foundation (Alzheimerfonden), Stiftelsen för Gamla Tjänarinnor, Gun and Bertil Stohnes Foundation, and the Swedish Research Council (2017-01874). Y Yu acknowledges support from the China Scholarship Council (201600160089) and Thomas Olausson. DC Jans acknowledges financial support for infrastructure development from the Swedish Foundation for Strategic Research (RIF14-0091). B Winblad acknowledges financial support from Margaretha af Ugglas' foundation and the Swedish Brain Foundation. Part of the study was performed at the Live Cell Imaging unit/Nikon Center of Excellence, Department of Biosciences and Nutrition, Karolinska Institutet, Huddinge, Sweden, supported by grants from the Knut and Alice Wallenberg Foundation, the Swedish Research Council, the Centre for Innovative Medicine, and the Jonasson donation to the School of Technology and Health, Royal Institute of Technology, Sweden. We thank Henrik Biverstål for kindly providing purified recombinant human Aβ42.

### Author Contributions

Y Yu: data curation, software, formal analysis, investigation, visualization, methodology, and writing—original draft, review, and editing.
DC Jans: software, formal analysis, investigation, visualization, methodology, and writing—original draft, review, and editing.
B Winblad: supervision and funding acquisition.
LO Tjernberg: conceptualization, resources, data curation, formal analysis, supervision, funding acquisition, validation, investigation, methodology, project administration, and writing—original draft, review, and editing.
S Schedin-Weiss: conceptualization, data curation, formal analysis, supervision, validation, investigation, visualization, methodology, project administration, and writing—original draft, review, and editing.

### Conflict of Interest Statement

The authors declare that they have no conflict of interest.

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
