## [Reviewer comments · Life Science Alliance]

Neuronal A β 42 is enriched in small vesicles at the presynaptic side of synapses

Yang Yu¹, Daniel C. Jans², Bengt Winblad¹, Lars Tjernberg¹ and Sophia Schedin-Weiss

DOI: 10.26508/lisa.201800028

Review timeline:	Submission date:	5 February 2018
	1 st Editorial Decision:	2 March 2018
	1 st Revision received:	15 May 2018
	2 nd Editorial Decision:	28 May 2018
	2 nd Revision received:	1 June 2018
	Accepted:	1 June 2018

Report:

(Note: Letters and reports are not edited. The original formatting of letters and referee reports may not be reflected in this compilation.)

1st Editorial Decision

2 March 2018

Thank you for submitting your manuscript entitled "Neuronal A β 42 is enriched in small vesicles at the presynaptic side of synapses" to Life Science Alliance. The manuscript was assessed by expert reviewers, whose comments are appended to this letter. We invite you to submit a revision if you can address the reviewers' key concerns, as outlined here.

As you will see, both reviewers point out the need for providing evidence for the specificity of the antibody used in the study and for a better discussion of the membrane association of A β 42. Even if the antibody specificity was previously demonstrated, the referees pointed out in our cross-commenting consultation session that it is important in the context of your study to show that the antibody is recognizing A β and not APP. Referee #1 furthermore requests further support for the distinction of the A β 42-positive vesicles from transmitter-releasing vesicles, and this request should get addressed as well when revising your work.

The requested amendments and controls seem straightforward in our view, but please get in touch should you want to discuss the revision further.

Thank you for this interesting contribution to Life Science Alliance. We are looking forward to receiving your revised manuscript.

REFeree REPORTS

Reviewer #1 (Comments to the Authors (Required)):

The authors use here antibody stainings to describe the organization of Abeta42, a pathogenic form of the amyloid beta peptide, in synapses.

The scope of the work is interesting, and the work is well performed.

However, before publication a few revisions are required:

- The authors mention that the Abeta42 antibody is specific for this molecule, with limited or no cross-reactivity to other species of APP or Abeta. Could this be tested by the authors directly, for example in some immunoblotting or similar experiments? Alternatively, could they reproduce some

of the stainings with a different Abeta42 antibody?

- The authors may want to test their claim that "these vesicles appeared to lack synaptophysin, indicating that they differ from synaptic vesicles and suggesting that A β 42 release at the synapse could be independent of neurotransmitter release" by immunostaining for two synaptic vesicle markers, such as synaptophysin and synaptotagmin, and testing their colocalization. Should the two bona fide vesicle markers colocalize far more strongly than Abeta and synaptophysin, then the claim is correct. Should the lack of colocalization be a caveat of the immunostaining technique, then this claim needs to be made less strong.

- The authors also claim that "A β 42 was instead present in the outer membrane of large, around 200-600 nm, vesicular structures". While the presence of the molecule in the outer membrane is expected (rather than on the inner membrane), this information cannot be derived from the images directly, so it may be better to simply state "membrane" in the abstract, rather than "outer membrane", and provide more details only in the Discussion.

Reviewer #2 (Comments to the Authors (Required)):

Dr. Yu and colleagues provide an interesting set of studies using state of the art imaging to show that Abeta42 is predominantly found within presynaptic terminals (97%!) and negligible amounts postsynaptically. Overall the studies are well-performed and described. That Abeta appears to be on the outside of vesicles is surprising since Abeta is made within endosomes. Adding some discussion as to how Abeta could translocate from the lumen of the vesicle to the outer leaf on the cytoplasmic side should be added to the manuscript. Additionally, Abeta can be found intracellularly and secreted. If Abeta42 is on the cytoplasmic side, how is it secreted from neurons normally or following synaptic activity? Or do the authors propose that this cytoplasmic Abeta is not part of the secreted pool? Some speculation would be interesting.

No primary antibody controls are included to demonstrate that the signal is antibody-dependent. But this does not prove that the antibody is specific to Abeta42 or even Abeta at all. Additional studies should be included to demonstrate that the signal here is not APP which is also at high levels in neurons. One way to demonstrate this would be to pre-treat cells for a long period of time with a gamma-secretase inhibitor to block Abeta production, then demonstrate the staining signal goes away.

1st Revision – authors' response

15 May 2018

Reviewer #1 (Comments to the Authors (Required)):

The authors use here antibody stainings to describe the organization of Abeta42, a pathogenic form of the amyloid beta peptide, in synapses. The scope of the work is interesting, and the work is well performed. However, before publication a few revisions are required:

- The authors mention that the Abeta42 antibody is specific for this molecule, with limited or no cross-reactivity to other species of APP or Abeta. Could this be tested by the authors directly, for example in some immunoblotting or similar experiments? Alternatively, could they reproduce some of the stainings with a different Abeta42 antibody?

Response

We used immunoblotting experiments (WB) to show that the A β 42 antibody is specific for the free C-terminus of A β 42, with no cross-reactivity with A β 40 or full-length APP (Fig EV1). Mouse 4G8 antibody, reactive to amino acid residues 17-24 of the A β sequence within APP, i.e. recognizing also full-length APP and C-terminal fragments, and the A β 42-specific antibody were tested on purified A β 42 and mouse brain homogenate samples. Please see more details on page 5 in Results and Discussion, page 11-12 in Materials and methods mice section, page 12-13 in Antibodies and reagents section, page 13-14 in the Western blotting section, and Expanded view figure legends for Fig EV1 on page 26.

- The authors may want to test their claim that "these vesicles appeared to lack synaptophysin, indicating that they differ from synaptic vesicles and suggesting that A β 42 release at the synapse could be independent of neurotransmitter release" by immunostaining for two synaptic vesicle markers, such as synaptophysin and synaptotagmin, and testing their colocalization. Should the two

bona fide vesicle markers colocalize far more strongly than Abeta and synaptophysin, then the claim is correct. Should the lack of colocalization be a caveat of the immunostaining technique, then this claim needs to be made less strong.

Response

To further investigate our observation that A β 42-containing vesicles lack synaptophysin, the colocalization between A β 42 and synaptophysin was compared with the staining of two synaptic vesicle proteins—synaptobrevin and synaptophysin (Fig EV4). The results showed that the synaptobrevin/synaptophysin colocalization was higher than the A β 42/synaptophysin colocalization. Please see more details on page 7-8 in Results and discussion, page 16-17 in Materials and methods and Expanded view figure legends for Fig EV4 on page 27. Since there was not 100% colocalization for the synaptobrevin/synaptophysin samples either, we have made or claim less strong in the abstract (page 2) and summary of the discussion (page 11).

- The authors also claim that "A β 42 was instead present in the outer membrane of large, around 200-600 nm, vesicular structures". While the presence of the molecule in the outer membrane is expected (rather than on the inner membrane), this information cannot be derived from the images directly, so it may be better to simply state "membrane" in the abstract, rather than "outer membrane", and provide more details only in the Discussion.

Response

We agree with the reviewers comment and have changed the text from "outer membrane" to "membrane". The discussion has also been changed accordingly on page 9-10.

Reviewer #2 (Comments to the Authors (Required)):

Dr. Yu and colleagues provide an interesting set of studies using state of the art imaging to show that Abeta42 is predominantly found within presynaptic terminals (97%!) and negligible amounts postsynaptically. Overall the studies are well-performed and described. That Abeta appears to be on the outside of vesicles is surprising since Abeta is made within endosomes. Adding some discussion as to how Abeta could translocate from the lumen of the vesicle to the outer leaf on the cytoplasmic side should be added to the manuscript. Additionally, Abeta can be found intracellularly and secreted. If Abeta42 is on the cytoplasmic side, how is it secreted from neurons normally or following synaptic activity? Or do the authors propose that this cytoplasmic Abeta is not part of the secreted pool? Some speculation would be interesting.

Response

We apologize if our wording in the previous manuscript version was confusing. We did not mean to say that A β 42 is present on the cytosolic side. We have therefore changed the wording and extended the discussion on page 9-10.

No primary antibody controls are included to demonstrate that the signal is antibody-dependent. But this does not prove that the antibody is specific to Abeta42 or even Abeta at all. Additional studies should be included to demonstrate that the signal here is not APP which is also at high levels in neurons. One way to demonstrate this would be to pre-treat cells for a long period of time with a gamma-secretase inhibitor to block Abeta production, then demonstrate the staining signal goes away.

Response

We used immunoblotting experiment (western blotting) to show the A β 42 antibody is specific for the free C-terminus of A β 42, with no cross-reactivity with APP (Fig EV1). Please see also the first response to reviewer 1 about the method and result.

In addition to the changes mentioned above, some minor changes have been done. A manuscript text version with all changes highlighted in red has been uploaded along with the other files in the submission portal. The page numbering above refers to that manuscript version.

Thank you for submitting your revised manuscript entitled "Neuronal A β 42 is enriched in small vesicles at the presynaptic side of synapses". Your manuscript has been seen by the original reviewers again, who both now support publication of your work.

REFeree REPORTS

Reviewer #1 (Comments to the Authors (Required)):

The authors have replied successfully to all of my comments, and I suggest that the manuscript be published.

Reviewer #2 (Comments to the Authors (Required)):

The authors have addressed all of my suggestions, including demonstrating that their Abeta42 antibody is highly selective for Abeta42 and does not bind to APP. I have no further comments.